

# Venomix: a simple bioinformatic pipeline for identifying and characterizing toxin gene candidates from transcriptomic data

Jason Macrander[1,2], Jyothirmayi Panda[3], Daniel Janies[3,4], Marymegan Daly[2] and Adam M. Reitzel[1]

[1] Department of Biological Sciences, University of North Carolina at Charlotte, Charlotte, NC, United States of America
[2] Department of Evolution, Ecology, and Organismal Biology, Ohio State University, Columbus, OH, United States of America
[3] College of Computing and Informatics, University of North Carolina at Charlotte, Charlotte, NC, United States of America
[4] Department of Bioinformatics and Genomics, University of North Carolina at Charlotte, Charlotte, NC, United States of America

## ABSTRACT

The advent of next-generation sequencing has resulted in transcriptome-based approaches to investigate functionally significant biological components in a variety of non-model organism. This has resulted in the area of "venomics": a rapidly growing field using combined transcriptomic and proteomic datasets to characterize toxin diversity in a variety of venomous taxa. Ultimately, the transcriptomic portion of these analyses follows very similar pathways after transcriptome assembly often including candidate toxin identification using BLAST, expression level screening, protein sequence alignment, gene tree reconstruction, and characterization of potential toxin function. Here we describe the Python package Venomix, which streamlines these processes using common bioinformatic tools along with ToxProt, a publicly available annotated database comprised of characterized venom proteins. In this study, we use the Venomix pipeline to characterize candidate venom diversity in four phylogenetically distinct organisms, a cone snail (Conidae; *Conus sponsalis*), a snake (Viperidae; *Echis coloratus*), an ant (Formicidae; *Tetramorium bicarinatum*), and a scorpion (Scorpionidae; *Urodacus yaschenkoi*). Data on these organisms were sampled from public databases, with each original analysis using different approaches for transcriptome assembly, toxin identification, or gene expression quantification. Venomix recovered numerically more candidate toxin transcripts for three of the four transcriptomes than the original analyses and identified new toxin candidates. In summary, we show that the Venomix package is a useful tool to identify and characterize the diversity of toxin-like transcripts derived from transcriptomic datasets. Venomix is available at: https://bitbucket.org/JasonMacrander/Venomix/.

Corresponding author
Jason Macrander,
jmacrand@uncc.edu

# INTRODUCTION

Throughout the animal kingdom, venom has evolved independently multiple times to be used in prey capture, predatory defense, and intraspecific competition (*Casewell et al., 2013*). Venoms are toxic cocktails with remarkable diversity in protein action and specificity across animals. The evolutionary and ecological processes shaping this diversity are of major interest (*Fry et al., 2009*; *Wong & Belov, 2012*; *Casewell, Huttley & Wüster, 2012*; *Sunagar et al., 2016*; *Rodríguez de la Vega & Giraud, 2016*), with much of this focusing on characterizing the composition of proteins and mRNAs expressed in the venom gland (*Ménez, Stöcklin & Mebs, 2006*). As sequencing costs decrease, the number of venom-focused studies is increasing at a dramatic rate. For some of the better studied venomous lineages (e.g., Colubroidea), comparative transcriptome and genome sequencing approaches are being used to investigate processes involved with toxin gene recruitment and tissue specific gene expression (*Vonk et al., 2013*; *Hargreaves et al., 2014*; *Reyes-Velasco et al., 2015*; *Junqueira-de Azevedo et al., 2015*). For other more poorly studied taxonomic lineages, similar techniques are being used to evaluate venom diversity using bioinformatic pipelines for a particular species or taxonomic group (*Tan, Khan & Brusic, 2003*; *Reumont et al., 2014*; *Macrander, Brugler & Daly, 2015*; *Kaas & Craik, 2015*; *Prashanth & Lewis, 2015*). Although these take similar approaches to study diverse venoms across animal lineages, a streamlined systematic pipeline does not exist for rapid identification of candidate toxin genes from transcriptomic datasets regardless of their taxonomic lineage.

Bioinformatic tools that use transcriptomic, proteomic, and genomic data sets have emerged for a variety venomous taxa. Among these, programs founded in machine learning appear to be the most abundant tools currently available; these use a combination of lineage specific annotation datasets (*Kaplan, Morpurgo & Linial, 2007*; *Fan et al., 2011*; *Wong et al., 2013*) and identifiers based on residue frequency and protein domains of interest (*Gupta et al., 2013*). Although taxonomic and tissue specific application of these programs vary, the pipelines follow a similar mechanical path (*Haney et al., 2014*; *Zhang et al., 2014*; *Júnior et al., 2016*; *Macrander, Broe & Daly, 2016*; *Durban et al., 2017*; *Verdes, Simpson & Holford, 2018*; *Rivera-de Torre, Martínez-del-Pozo & Garb, 2018*). First, they begin with millions of raw reads assembled *de novo* using Trinity (*Grabherr et al., 2011*) or similar (*Archer et al., 2014*). Next, expression values for each transcript are calculated using RSEM (*Li & Dewey, 2011*) or similar programs, like BWA (*Li & Durbin, 2009*). Ultimately the resulting transcriptome assembly is searched for toxin candidates by some component of BLAST (*Camacho et al., 2009*; *Neumann, Kumar & Shalchian-Tabrizi, 2014*) or other motif-searching algorithms (*Kozlov & Grishin, 2011*; *Finn, Clements & Eddy, 2011*). For many of the pipelines, these outputs are then screened using custom query datasets comprised of lineage specific toxin genes (*Tan et al., 2006*; *Kaas et al., 2012*; *Pineda et al., 2018*) or the manually curated ToxProt dataset (*Jungo et al., 2012*), which includes all characterized/annotated animal venom proteins. Following candidate toxin gene identification, downstream analyses often involve predicting open-reading frames using Transdecoder (*Haas et al., 2013*) or similar, in combination with signal region prediction

using SignalP (*Petersen et al., 2011*). These types of data are sometimes complemented with genome and proteome datasets (see *Sunagar et al., 2016*). However, the majority of studies that are exploratory in nature use transcriptomic approaches to describe overall toxin diversity for a variety of poorly studied taxa (*Reumont et al., 2014*; *Macrander, Brugler & Daly, 2015*; *Barghi et al., 2015*; *Luna-Ramírez et al., 2015*; *Macrander, Broe & Daly, 2016*; *Lewis Ames et al., 2016*). One major drawback to this approach, and using these self-constructed pipelines, is that downstream analyses begin to become quite cumbersome when trying to identify and characterize multiple toxin gene families for diverse toxin genes found within a large transcriptome.

Here we present Venomix, a bioinformatic pipeline written in the programming languages Python and R that follows commonly used methods for identifying and characterizing toxin-like genes from transcriptomic datasets. In this study, we use Venomix to characterize the toxin-like diversity from venom gland transcriptomes for a cone snail (*Conus sponsalis*), a snake (*Echis coloratus*), an ant (*Tetramorium bicarinatum*), and a scorpion (*Urodacus yaschenkoi*). Venomix incorporates widely used programs into its pipeline, including BLAST (*Camacho et al., 2009*) for initial toxin-like transcript identification; Transdecoder (*Haas et al., 2013*) to translate transcripts into their proper reading frame, SignalP (*Petersen et al., 2011*) to predict toxin gene signaling regions, MAFFT (*Katoh & Standley, 2013*) for protein sequence alignment, and the R package APE (*Paradis, Claude & Strimmer, 2004*) to construct gene trees. Candidate toxin genes are grouped based on sequence similarity, with each directory corresponding to a specific toxin group based on the ToxProt sequence names (e.g., some variation of conotoxin, Kunitz-type serine protease, phospholipase A2, zinc metalloproteinase). The Venomix pipeline provides the user with several output files that can be used to characterize the potential function of these candidate toxins, compare relevant expression level values across toxin-gene candidates, evaluate amino acid conservation among functionally important residues in sequence alignments, and review taxonomic and functional information in combination with tree reconstructions to further evaluate toxin gene candidates. Although Venomix is not meant to be a definitive validation pipeline for toxin genes, it can quickly identify, sort, and characterize transcripts that may be used to further evaluate these toxin candidates. By abating time required by these processes, researchers can then focus on downstream proteomic or functional analyses to better understand venom diversity present in the transcriptome.

# MATERIALS AND METHODS

## Data acquisition and transcriptome assembly

Raw reads from four different analyses were downloaded from the short read archive (SRA) on GenBank (*C. sponsalis*: SRR260951 (*Phuong, Mahardika & Alfaro, 2016*); *U. yaschenkoi*: SRR1557168 (*Luna-Ramírez et al., 2015*); *E. colaratus*: ERR216311–ERR216312 (*Hargreaves et al., 2014*); *T. bicarinatum*: SRR1106144–SRR1106145 (*Bouzid et al., 2014*)). The previously published transcriptome level analysis for *U. yaschenkoi* and *T. bicarinatum* were restricted to only characterizing the venom gland transcriptome in their
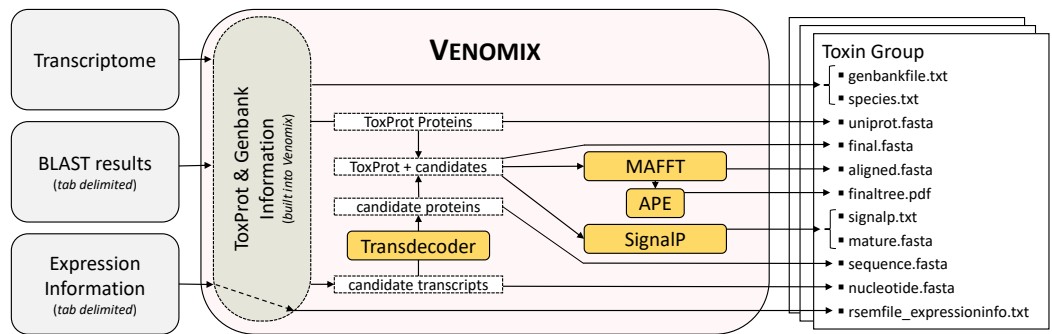

**Figure 1** **Venomix Pipeline Outline.** Outline showing the stepwise progression of the Venomix pipeline, including the necessary inputs and files included for every Toxin Group directory. Orange boxes indiciate programs used.

respective species (*Bouzid et al., 2014*; *Luna-Ramírez et al., 2015*). While the *C. sponsalis* and *E. coloratus* venom gland transcriptomes were investigated in a comparative framework alongside closely related taxa (*Hargreaves et al., 2014*; *Phuong, Mahardika & Alfaro, 2016*). In our study, all four transcriptomes were assembled in Trinity (*Grabherr et al., 2011*; *Haas et al., 2013*), using default parameters of its built-in Trimmomatic program to clean up the sequences (*Bolger, Lohse & Usadel, 2014*). For each transcriptome, expression values as transcripts per kilobase million (TPM) and fragments per kilobase million (FKPM) were calculated in the program RSEM (*Li & Dewey, 2011*) as part of the Trinity package.

## Analysis pipeline and execution

The bioinformatic pipeline for Venomix is outlined in Fig. 1. The program requires three inputs provided by the user: (1) an assembled transcriptome, (2) gene expression information in the form of a tab-delimited output with transcript names in the first column, and (3) tab-delimited BLAST output using the ToxProt as query sequences against the trainscriptome. Although we used Trinity and RSEM in our study, most assembly and expression profile programs with these formats should work with Venomix. Following transcriptome assembly and expression level calculations, the final user provided file is created using tBLASTn from NCBI BLAST+ version 2.4.0 (*Camacho et al., 2009*), with the ToxProt dataset as the search query with the final BLAST alignment results shown in a tabular format (-outfmt 6). Query sequences from ToxProt are provided within the Venomix package; however, alternative curations of the ToxProt dataset may be used if the sequence identifiers are not changed. In our analysis, we implemented two BLAST search procedures; the first used a more stringent identification algorithm (*E*-value 1e−20) and a less stringent identification algorithm (*E*-value 1e−6).

Venomix was tested on the University of North Carolina at Charlotte COPPERHEAD Research Computing Cluster, while requesting computational resources that may mimic most personal laptops/desktops, specifically one processor and 4 GB of RAM. Using these settings the Venomix completed in less than twenty minutes for each of the focal transcriptome. The implementation of Venomix requires the scripting

languages Python 2.7 (http://www.python.org/download/releases/27/) and R 3.1.1 (https://cran.r-project.org/bin/), in addition to other Biopython packages (*Cock et al., 2009*) and data from ToxProt and Genbank that are built into the Venomix pipeline (https://bitbucket.org/JasonMacrander/venomix). We included versions of MAFFT (*Katoh & Standley, 2013*), NCBI BLAST+ (*Camacho et al., 2009*), and Transdecoder (*Haas et al., 2013*) that can be run locally. Although there are two versions of MAFFT (64 bit and 32 bit), the default is the 64-bit, as this is more common for computers used in bioinformatic analyses. Modification to the version of MAFFT in the Venomix pipeline can be done in the support_files/Alignment.py file.

Once the user specifies the input files (transcriptome, expression file, and BLAST output), the Venomix pipeline automatically produces several potentially useful and informative files within each of the toxin group directories (Fig. 1). The outputs within each of the Toxin Group directories are intended to provide the user with curated information to focus future investigations and analyses. Specifically the outputs are as follows: (1) unaligned full and partial nucleotide sequences representing candidate toxins, (2) UniProt query sequences used to identify candidates, (3) taxonomic information associated with query sequences, (4) Genbank entries from query sequences, (5) translated protein sequences from full nucleotide sequences (no partial), (6) UniProt toxin and candidate protein sequences unaligned and (7) in alignment, (8) SignalP information, (9) mature peptides when signaling regions are present, (10) a neighbor joining tree based on the protein sequence alignment, and (11) expression information, if provided by the user. All of these files are contained in directories with representative toxin names retrieved from ToxProt proteins. As the pipeline does not definitively identify candidate toxins to toxin type, similar toxin gene families likely have similar candidates in different directories.

Depending on the next step of the analysis, some of the output files may be used for additional venom related downstream applications or simply a quick reference for the user. Venomix also creates two ancillary products that may be informative to some users: TPM.fasta (only transcripts with TPM values >1.0) and a large GenBank file with information from ToxProt BLAST hits in a format that may lend itself to quick searches or downstream annotation. The user may choose to rerun Venomix with TPM.fasta instead of their assembled transcriptome if they would like to characterize only transcripts with a TPM >1.0, but this is not recommended when looking for rare or extremely diverse toxin genes.

## Venomix evaluation

For each assembled transcriptome, we identified candidate toxin genes using the Venomix pipeline using a stringent ($E$-value 1e−20) and less stringent ($E$-value 1e−6) search strategy in BLAST. Venomix outputs were compared for both search parameters in terms of the number of toxin groups, number of transcripts, and number of "candidate" transcripts identified by the pipeline. A transcript was considered a "candidate" if the transcript had significantly better $E$-value associated with a toxin than with a non-toxin protein in Uniprot. Candidate transcripts were translated into their protein sequences using Transdecoder (*Haas et al., 2013*) and further evaluated in ToxClassifier (*Gacesa, Barlow &*

*Long, 2016*). If a predicted protein sequence received a score of >1 it was considered a toxin candidate. In addition to screening the overall transcriptome analyses, some toxin groups and candidate toxins identified in our analysis were subjected to additional screening beyond what is included in the Venomix pipeline representative of what some users may do in downstream analyses. Sequence alignments for candidate transcripts shown below were done using MAFFT (v.1.3.3) (*Katoh & Standley, 2013*) and visualized in Geneious (*Kearse et al., 2012*). Toxin gene tree reconstructions were done in Fasttree v2 (*Price, Dehal & Arkin, 2010*) using maximum likelihood tree reconstruction methods and bootstrap supports calculated over 1,000 replicates. For the *Bouzid et al. (2014)* dataset, Venomix was used to compare alternative assembly approaches (Oases/Velvet vs. Trinity), in addition to assessing both transcriptomes for overall completeness in BUSCO (*Simão et al., 2015*). Expression values for each transcriptome were calculated using RSEM (*Li & Dewey, 2011*) rather than raw read counts as originally proposed by *Bouzid et al. (2014)*.

# RESULTS

## Transcriptome assemblies

Transcriptomes reassembled in Trinity (*Grabherr et al., 2011*) produced similar *de novo* assembly outputs when compared to the original studies (Table S1), with the only notable difference being in the number of transcripts for *C. sponsalis,* which may be due to repetitiveness and sequence complexity encountered during their initial assemblies (*Phuong, Mahardika & Alfaro, 2016*). The transcriptome for *T. bicarinatum* was originally done using Velvet/Oases (*Li & Durbin, 2009*); however, we compared this to our Trinity assembly because of its ease of use (*Sanders et al., 2018*) and frequency in the venom literature (*Macrander, Broe & Daly, 2015*), in addition to a lower redundancy and chimera rate (*Yang & Smith, 2013*).

## Pipeline output

In the original published studies, species-specific transcriptomes of *C. sponsalis*, *E. colaratus* and *U. yaschenkoi* were not subjected to any BLAST searches using ToxProt, but instead were screened using taxonomic specific toxin datasets from venom proteins of closely related species. The Venomix pipeline recovered the majority of these lineage specific toxins, as well as additional transcripts that resemble toxin genes from other taxa. It is worth noting that if there were lineage specific toxins that shared high sequence similarity to other toxins from distantly related taxonomic groups, the toxin group name may be assigned an incorrect lineage classification, yet remain a toxin candidate. For example, analyses of the scorpion *U. yaschenkoi* resulted in four venom groups with "Snake venom" in the name; however, in these instances, lineage specific toxin names are often members of larger gene families that may not be lineage specific toxins. The number of identified toxin groups varied considerably across species and stringency parameters (Table 1), with the less stringent parameters dramatically increasing the number of toxin groups. Each species had multiple toxin groups that were separated based on sequence similarity and correspond to large gene families, including astacin-like metalloproteases, conotoxins, phospholipases A2s, CRISPs, Kunitz-type serine protease inhibitors, snaclecs,

**Table 1** Species-specific Venomix outputs following different search strategies

| | | Original publication | | | Stringent (*e-value 1e-20*) | | | Less stringent (*e-value 1e-6*) | | |
|---|---|---|---|---|---|---|---|---|---|---|
| | *N* | Groups | Transcripts | Evaluated | Groups | Transcripts | Evaluated | Groups | Transcripts | Evaluated |
| *C. sponsalis* | *401* | 35[c] | 780[†] | 393(363) | 22 | 61 | 44(13) | 75 | 293 | 246(45) |
| *E. coloratus* | *34* | 8 | 82 | 62(35) | 72 | 339 | 147(116) | 130 | 775 | 202(143) |
| *T. bicarinatum* | *69* | 32 | 527[β] | 287/62[E](14) | 36 | 289 | 95(14) | 75 | 761 | 201(36) |
| *U. yaschenkoi* | *111* | 11[s] | 210 | 71(6) | 50 | 277 | 48(34) | 117 | 689 | 179(38) |

**Notes.**

N, number of candidate toxins identified in original study; Groups, number of toxins types identified based on sequence similarity; [c], conotoxins only (*Phuong, Mahardika & Alfaro, 2016*); [s], scorpion toxins only (*Luna-Ramírez et al., 2015*); Transcripts, total number of unique transcripts evaluated; [†], includes duplicates as cumulative after three iterations in Trinity [see *Phuong, Mahardika & Alfaro, 2016*]; [β], >100 TPM difference upregulated in the venom gland compared the ant carcass; [E], number of candidates based on different *E*-values 10/1E−3 thresholds; Evaluated, number of unique transcripts retained after using BLAST screening, parenthesis indicates number of transcripts identified using a Toxclassifier score of 1 or greater.

metalloproteinases, thrombin-like enzymes, and allergens. For each species, there was at least one toxin group that was not retained following the reciprocal BLASTp search after the toxin-like transcripts were translated into their open reading frame (Table 1).

## Venomix outputs for *C. sponsalis*

Conotoxins represent some of the best-studied toxin genes across the genus *Conus*, comprising of multiple gene families with cysteine rich proteins (*Buczek, Bulaj & Olivera, 2005*; *Kaas et al., 2012*). Venomix identified 76 toxin groups comprising of 246 toxin gene candidates based on our preliminary low stringency BLAST survey; 20 of these groups cluster with various conotoxins (Fig. 2). In total, there were 179 of the 246 toxin gene candidates from the 20 conotoxin groups. There were three instances where Venomix recovered more candidate conotoxins than the original study (O1, conodipine, and conophysin superfamilies), however, a majority of the conotoxin superfamilies identified by *Phuong, Mahardika & Alfaro (2016)* were missing from our analysis (Fig. 2). This discrepancy is likely due to the different assembly approaches, as iterative assemblies used by *Phuong, Mahardika & Alfaro (2016)* were unable to recover known transcripts using Trinity alone. Beyond the conotoxins, there were 19 candidate toxin genes found within the Kunitz-type conkuitzin-S1 group, which included characterized toxin proteins from the venom Kunitz-type family of sea anemones, cone snails, and snakes (Supplemental Information 1).

## Venomix outputs for *E. coloratus*

Snakes represent one of the best-studied animal lineages and we were able to validate Venomix's efficiency with the *E. coloratus* transcriptome. With a low stringency search (*E*-value = 1E−6) Venomix identified 132 toxin groups (Table S2), Among the transcripts identified, 45 had a TPM value greater than 100, with 39 of these in the venom gland, four in the scent gland, and two in the skin. The majority of the highly expressed transcripts in the venom gland (TPM > 100) corresponded with toxin groups previously identified (*Hargreaves et al., 2014*), comprising mostly of C-type lectins, cysteine rich venom proteins, disintegrins, metalloproteinases, and several others (Fig. 3). In addition to these venom candidates, we found one highly expressed cystatin in the venom gland, although it was

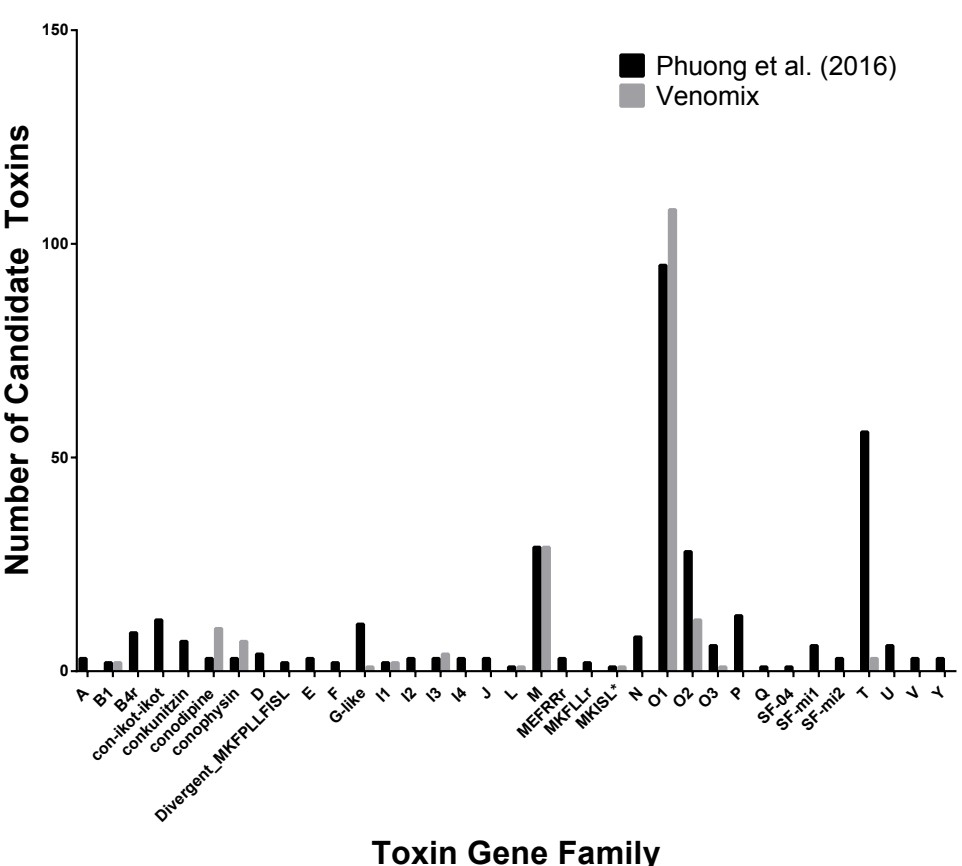

**Figure 2** **Comparison of Conotoxin Transcripts for *C. sponsalis*.** Number of candidate toxin transcripts from each toxin gene family from the original study (*Phuong, Mahardika & Alfaro, 2016*) and Venomix.

also highly expressed in other tissues and not thought to be a toxic component of the *E. coloratus* venom (*Hargreaves et al., 2014*). We also identified a peroxiredoxin, ficolin, and three latroinsectotoxins (Supplemental Information 2), all of which may have some role during envenomation now well characterized in snakes (*Magazanik et al., 1992*; *Calvete et al., 2009*; *OmPraba et al., 2010*), but require further investigation.

## Venomix outputs for *T. bicarinatum*

Ants represent a poorly-studied lineage of venomous animals, with *T. bicarinatum* being the only transcriptome used in our study which identified toxins using the ToxProt dataset (*Bouzid et al., 2014*). The original transcriptome assembly was done in Velvet/Oases (*Li & Durbin, 2009*). Despite the alternative approaches, the BUSCO (*Simão et al., 2015*) scores were similar, with the Velvet/Oases assembly at 95.9% and Trinity at 92.2%. When considering TPM (rather than raw counts) the number of candidate genes in the venom gland following the approach by *Bouzid et al. (2014)* were similar to what was originally published (Table 1). Among these 527 candidates, there were only 44 predicted ORFs from Transdecoder, and only three of these were given a score of 1 or greater in Toxclassifier (Table 1). The BLAST screening, however, resulted in 62 candidate toxins identified when

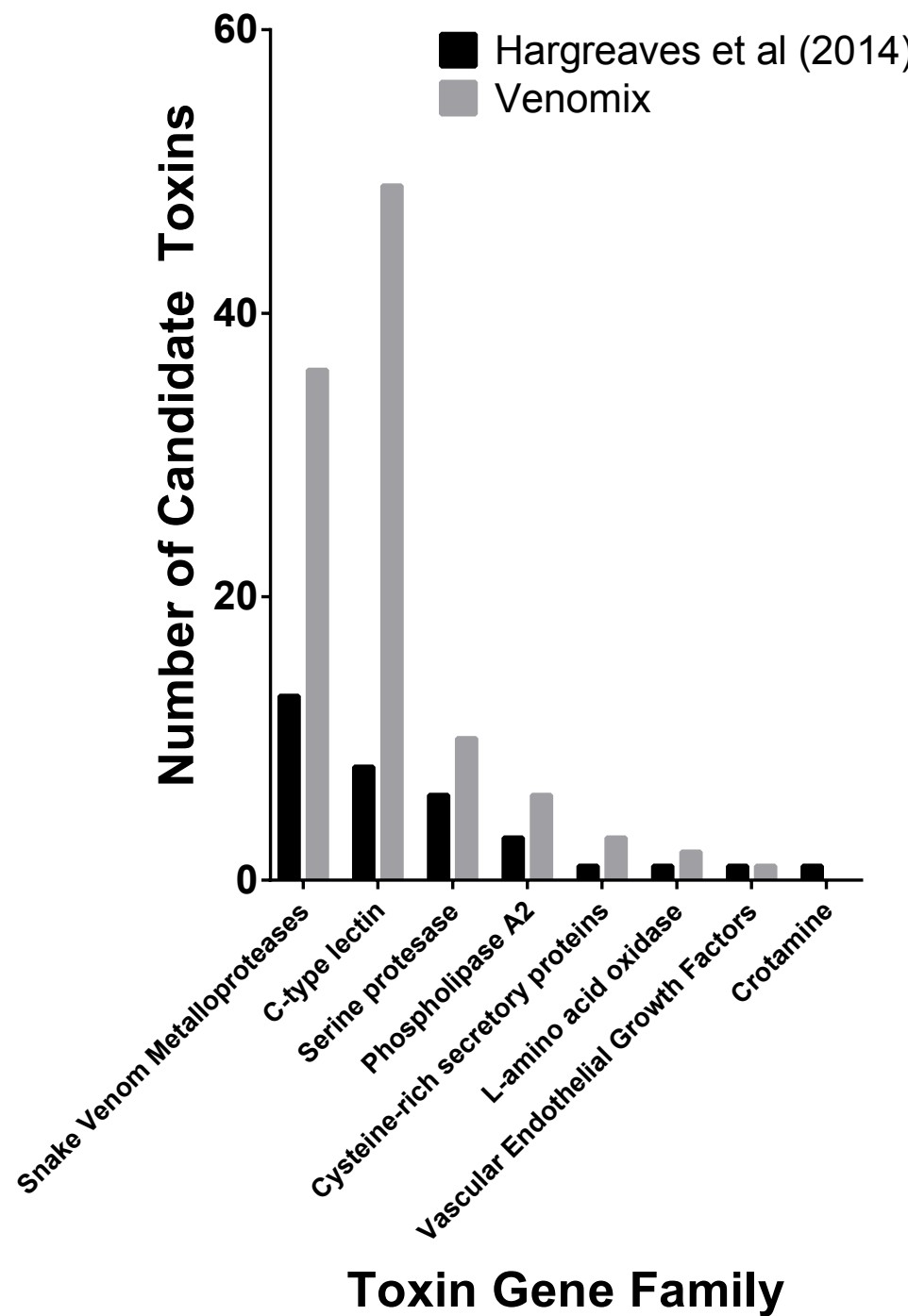

**Figure 3** **Number of previously predicted toxin compared to those derived from Venomix.** Number of candidate toxin transcripts from each toxin gene family from the original study (*Hargreaves et al., 2014*) and Venomix candidates most highly expressed in the venom gland with a TPM >1.0.

the *E*-value threshold was set to 1E−3, but 287 when the *E*-value threshold was set to 10. As *E*-values were not specified by *Bouzid et al. (2014)* both are reported here (Table 1). In the less stringent analysis, the largest number candidate toxin genes corresponded to 280 transcripts in the alpha-latroinsectotoxin-Lt1a group, but overall expression within the ant carcass and venom gland across these transcripts were approximately the same. Among those more highly expressed transcripts in the venom gland, six had TPM scores greater than 100 in the venom gland; four corresponding to Venom Allergen 3 and two to cysteine-rich venom protein Mr30. Subsequent BLAST searches against UniProt indicated that all six of these toxins are likely Venom Allergen 3 toxins, making up ~92% of the cumulative toxin genes expressed within the transcriptome (Supplemental Information 3).

When we compared the Venomix outputs for our Trinity assembly to the Velvet/Oases assembly that was previously published by *Bouzid et al. (2014)* we recovered some unexpected results. Although we used RSEM instead of BWA, only 33 of the original 69 candidate toxin sequences recovered from their analysis had higher TPM expression in the venom gland than in the ant carcass. Of these 33 potential toxins, only 10 had a 1,000 fold higher expression based on expected count values (Table S3). Such a drastic difference in the reported fold change cross these toxins were unexpected when considering only the mapping program was different. Further examination revealed that thrombin-like enzymes from the Velvet/Oases assembly were likely only partial sequences (Supplemental Information 4), which may have contributed to the larger than 50% discrepancy between our analyses and is likely the reason why only 47 of the 8,688 transcripts identified in Venomix were translated into their open reading frame.

## Venomix outputs for *U. yaschenkoi*

Scorpions represent one of the oldest and best-studied venomous lineages, with the original study by (*Luna-Ramírez et al., 2015*) using scorpion-specific toxins as query sequences. Ultimately this original study identified 210 transcripts representing 111 unique scorpion toxins, venom gland enzymes, and antimicrobial peptides (*Luna-Ramírez et al., 2015*). By expanding the query sequences with the ToxProt dataset, we recovered 117 toxin groups representing 689 unique transcripts (Table 1). Of the 117 identified toxin groups only 10 were from scorpion derived query sequences. Of these 10, the Toxin-like protein 14 had the same number of candidate toxins as the original study, with Venomix recovering the complete protein sequence, when *Luna-Ramírez et al. (2015)* did not (Fig. 4A). When using exclusively scorpion venom proteins from ToxProt as query sequences, the number of candidate toxins identified by Venomix was approximately the same as that identified by *Luna-Ramírez et al. (2015)*.

The most abundant toxin-like transcripts within the less stringent search found 190 transcripts within the delta-latroinsectotoxin-Lt1a group, the 182 transcripts alpha latroinsectotoxin-Lt1a group, and 91 transcripts within the Neprilysin-1 group (Supplemental Information 5). Query toxins which are used to form these toxin groups were previously identified in spiders (*Graudins et al., 2012*; *Garb & Hayashi, 2013*; *Undheim et al., 2013*; *Bhere et al., 2014*), and not included in the original transcriptome analysis (*Luna-Ramírez et al., 2015*). Preliminary screening based on reciprocal BLAST hits indicated

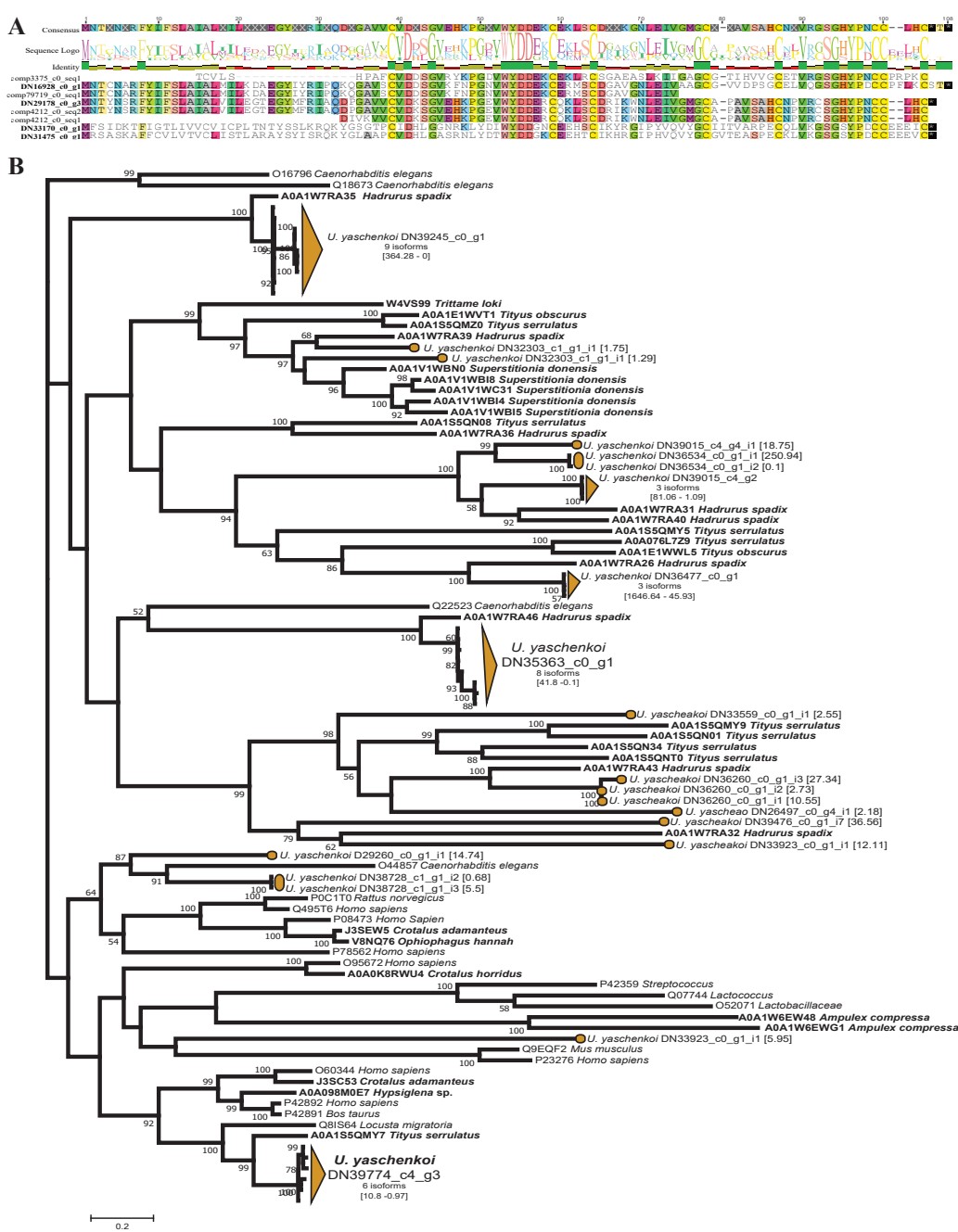

**Figure 4 Candidate toxins from *U. yaschenkoi*.** (A) Candidate toxins from *U. yaschenkoi* highlighting alignment difference in the candidate Toxin-like protein 14 sequencing in both analyses. Conserved residues across the alignment are highlighted. (B) Maximum likelihood neprilysin gene tree highlighting the abundance and diversity of candidate neprilysin toxins and non-venomous neprilysin genes. Branches associated with transcripts from *U. yaschenkoi* are highlighted in orange throughout the tree. Venomous taxa emphasized with bold font.

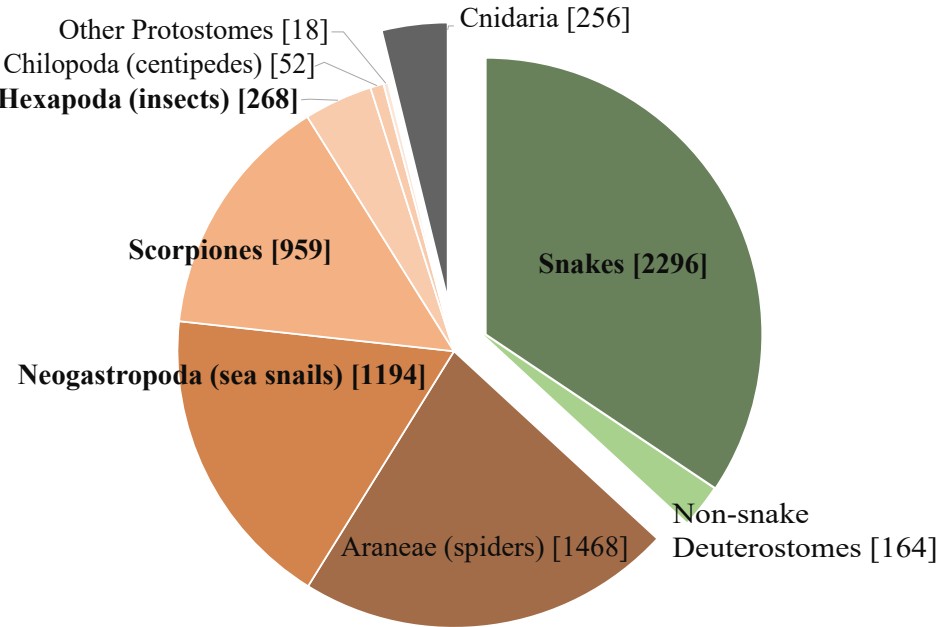

**Figure 5** **Taxonomic distribution of venom and toxin proteins in the ToxProt dataset.** Deuterostomes are highlighted in green, protostomes in brown, and cnidarians in grey.

that the latroinsectotoxin groups identified in Venomix might not be toxins. Maximum likelihood gene tree reconstructions were used as post-processing steps to further screen the Neprilysin-1 group as potential toxin sequences revealing that candidate toxins from the Neprilysin-1 group formed a well-supported cluster with neprilysin toxins from other scorpions at high expression levels (Fig. 4B).

## DISCUSSION

Venomix presents a less cumbersome, non-taxon specific alternative to any other pipeline currently being implemented in venom research. The pipeline allows the user to quickly identify and characterize toxin gene candidates within a transcriptomic dataset. The outputs provided by this pipeline give necessary information for further evaluation of their toxin gene candidates when complemented with proteomic or other approaches. We recommend using Venomix with multiple BLAST searches with varying $E$-value thresholds as the variation among characterized toxin genes and those of the focal taxa may be more accommodating depending on the threshold used. Although Venomix was able to identify more candidate toxin genes in three out of the four datasets tested here, these results require further examination to determine which transcripts are viable toxin gene candidates. Venomix is not intended to be a definitive toxin gene identifier because this determination should not be made by sequence data alone, especially for poorly studied lineages.

We chose four very different studies to highlight some of the benefits and limitations of Venomix. Of the taxa used in this study, three of them are from taxonomic groups with ample representation in the ToxProt dataset (Fig. 5), whereas the ant venom is poorly

characterized among the diverse venomous insects found within Hexapoda on ToxProt. Additionally, these datasets represent diverse transcriptome assembly methods, query datasets, and gene expression quantification approaches. Depending on the focal venom transcriptome, the assembly method may have a significant impact on how well toxin genes are recovered (*Holding et al., 2018*). The original *C. sponsalis* assembly had a high number of toxin genes with relatively low variation across gene copies (*Phuong, Mahardika & Alfaro, 2016*), which likely resulted in many of these being clumped together in our Trinity assembly (*Macrander, Broe & Daly, 2015*). To get around this issue, *Phuong, Mahardika & Alfaro (2016)* did three assembly iterations involving toxin gene identification and subsequent mapping, in addition to downstream analysis incorporating the Assembly by Reduced Complexity pipeline (https://github.com/ibest/ARC) and manual alignments in Geneious (Biomatters, Auckland, New Zealand). In contrast to *C. sponsalis*, the differences we observed in *T. bicarinatum* using Venomix was likely due to the alternative transcriptome assembly and gene expression approaches (*Yang & Smith, 2013*; *Vijay et al., 2013*; *Todd, Black & Gemmell, 2016*). Finally, limiting query sequences to only venoms of that lineage—which was done with the *C. sponsalis*, *E. coloratus*, and *U. yaschenkoi,* but not for *T. bicarinatum*—likely limited the number of toxin candidates being identified.

Despite having multiple assembly or expression approaches currently available for comparative venom transcriptome studies, we chose Trinity and RSEM because they are widely used in these types of analyses (*Haney et al., 2014*; *Zhang et al., 2014*; *Júnior et al., 2016*; *Macrander, Broe & Daly, 2016*; *Durban et al., 2017*; *Verdes, Simpson & Holford, 2018*; *Rivera-de Torre, Martínez-del-Pozo & Garb, 2018*). Venomix is capable of producing similar outputs regardless of assembly or quantification program when the input files are formatted properly. The Venomix pipeline was designed to sidestep much of the rigorous analysis used to identify and extract candidate toxin sequences, but is limited when considering what is actually translated into proteins. Our pipeline will translate full nucleotide transcripts into their predicted proteins, screen for signaling regions, assess their similarity through alignment and gene trees, and extract expression information. Complementing this analysis with proteomic datasets will allow users to identifying toxins in venoms using existing complementary protein specific tools through HMMs searches (*Finn, Clements & Eddy, 2011*) or other protein specific analyses. Venomix is the first pipeline to provide all these outputs in an easy to use search strategy that is flexible, but also repeatable, for all venomous taxa, or non-venomous animal to be used in a tissue specific comparative context (*Reumont et al., 2014*; *Hargreaves et al., 2014*; *Reyes-Velasco et al., 2015*). The pipeline also provides users with easy to navigate directories and organized output files (see Supplemental Informations 1–5), allowing the user to sort manually or quickly pull information for all toxin groups using simple Unix commands (i.e., grep) as the files within each toxin group directory have the same name.

Venomix can facilitate the process of determining what constitutes a venom protein and aid in testing future hypotheses of venom diversity and tissue specific expression. The *E. coloratus* transcriptome used in our analysis was part of a broader study to test the early evolution of venom in reptiles, the Toxicofera hypothesis (*Hargreaves et al., 2014*). They used tissue specific expression in combination with toxin gene tree reconstruction

to determine which of the approximately 16 venom toxin gene families that occur across Toxicofera attribute to the *E. coloratus* venom transcriptome. Using Venomix, toxin candidates can be identified using a similar approach as outputs within each toxin group contain gene expression information allowing for tissue specific comparisons with ease. Beyond venom candidates identified by *Hargreaves et al. (2014)*, there are also transcripts highly expressed in the venom gland that are likely not venomous (*Terrat & Ducancel, 2013*). This was made evident in the *U. yaschenkoi* analysis, as several transcripts within the latroinsectotoxins cluster were actually neprilysins in high abundance, but transcripts resembling neprilysins matched to other neprilysin toxins in a reciprocal BLAST hit.

Regardless of the bioinformatic approach to identifying toxin genes, one major hurdle using these types of datasets as query sequences is the limited taxonomic diversity present in the ToxProt dataset. Although the transcriptome for *U. yaschenkoi* was larger and had a longer N50 than that of *E. coloratus* (Table S1), there were more toxin-like transcripts identified in the *E. coloratus* transcriptome. This likely reflects the abundance of snake proteins deposited into ToxProt and is in contrast to the paucity of proteins for other, poorly studied venomous lineages (Fig. 5). Additionally, Venomix "group" names should be examined closely because some candidate toxin genes were labeled with lineage-specific proteins. For example, our analysis recovered a group called conophysin (a cone snail toxin) for *T. bicarinatum*, however, the transcripts associated with this appeared to be neurophysins. This was also observed when Venomix failed to group Venom Allergen 3 and Cysteine-rich venom protein Mr30 groups together for *T. bicarinatum*, even though it was apparent that the most highly expressed were all Venom Allergen 3 genes. When investigating venom diversity for poorly studied taxa, caution is warranted in using these gene names because the specific classifiers of the Venomix outputs provide a starting point for toxin gene identification but does not act as a distinct classification system.

In every transcriptome, the machine-learning program ToxClassifier failed to recover all of the toxins identified in their respective publications (Table 1). Our downstream analysis of the protein sequences produced by TransDecoder included any candidate toxin with a score >1, which severely over represents candidate toxins, as ToxClassifier considers a "potential toxin" score >4. Ultimately this would reduce the number of "toxins" for *C. sponsalis*, dropping it to 243 and *E. coloratus* to seven. Despite this, one major contrast between ToxClassifier and Venomix is that our pipeline is not meant to be a toxin gene identifier. Venomix was designed to be useful for preliminary searches for users new to the command line, or provide a platform that is adaptable for those that are well versed in the command line. The incorporated alignment and tree building methods are rudimentary and meant to be used for only initial screenings. This allows users to focus their efforts on downstream analyses using complementary proteomics and machine learning to differentiate between functionally toxic and non-toxic venom components (*Gacesa, Barlow & Long, 2016*) or to complement their transcriptomic data with functional assays of proteins or crude venom extracts.

## ACKNOWLEDGEMENTS

Venomix is a direct byproduct of the training received at Friday Harbor Marine Labs while attending the Practical Computing for Biologists workshop with Steve Haddock and Casey Dunn. The authors thank them and other members of the workshop (Aurturo Alvarez-Aguilar, Jimmy Bernot, Bill Browne, Anela Choy, Zander Fodor, Michelle Gather, Joel Kingslover, Jasmine Mah, Adelaide Rhodes, Liz Scheimer, Emily Warschefsky, Linda Wordeman, and Sara Wyckoff) for their continued support and enthusiasm for Venomix. The authors would also like to thank Edwin Rice for preliminary input on Python scripts, Daft Punk composing good music to allow for coding, and Tyler Carrier for his PeerJ enthusiasm.

### Funding

Support for this project was provided by NSF DEB 1257796 to Marymegan Daly, NSF DEB 1536530 to Marymegan Daly and Jason Macrander, and NSF OCE 1536530 to Adam Reitzel. Jyothirmayi Panda was supported by a research assistantship provided by the University of North Carolina at Charlotte Graduate School under the guidance of Dan Janies. The funders had no role in study design, data collection and analysis, decision to publish, or preparation of the manuscript.

### Grant Disclosures

The following grant information was disclosed by the authors:
NSF: DEB 1257796, DEB 1536530, OCE 1536530.
University of North Carolina at Charlotte Graduate School.

### Competing Interests

The authors declare there are no competing interests.

### Author Contributions

- Jason Macrander conceived and designed the experiments, performed the experiments, analyzed the data, prepared figures and/or tables, authored or reviewed drafts of the paper, approved the final draft.
- Jyothirmayi Panda conceived and designed the experiments, performed the experiments, approved the final draft.
- Daniel Janies and Adam M. Reitzel contributed reagents/materials/analysis tools, authored or reviewed drafts of the paper, approved the final draft.
- Marymegan Daly conceived and designed the experiments, authored or reviewed drafts of the paper, approved the final draft.

### Data Availability

Venomix Pipeline: https://bitbucket.org/JasonMacrander/venomix.

## Supplemental Information

Supplemental information for this article can be found online at http://dx.doi.org/10.7717/peerj.5361#supplemental-information.

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
