# Peer review of "Venomix: a simple bioinformatic pipeline for identifying and characterizing toxin gene candidates from transcriptomic data"

_PeerJ, doi:10.7717/peerj.5361_

## Round 0.1 · original submission · Major Revisions

Dear Dr. Macrander and colleagues:

Thanks for submitting your manuscript to PeerJ. I have now received three independent reviews of your work, and as you will see, two reviewers raised many concerns about the research. I am particularly concerned with the four untested claims identified by reviewer 1 and agree with the reviewer that these need to be robustly addressed. More minor concerns involve reported novelty, experimental design, data collection and presentation (particularly clarity), and explanation of protocols and approaches. These also need to be addressed in your revision. Another concern is whether or not non-experts can truly benefit from Venomix. Is this another tool that really only assists bioinformaticians or can bench biologists benefit from it? Can it be run on a laptop or standard desktop computer? Reviewer 2 raises concerns over statistics that need to be addressed. There appear to be missing references, missing software that should be integrated into Venomix, as well as several areas of the manuscript where English can be improved (clarity as well).

I am recommending that you revise your manuscript accordingly, taking into account all of the issues raised by the reviewers. Please keep in mind that your work should specifically address what distinguishes Venomix from other approaches and how it will help a wide range of biologists in their research on toxins..

I look forward to seeing your revision, and thanks again for submitting your work to PeerJ.

Good luck with your revision,

-joe

Reviewer 1 ·

Basic reporting

The manuscript is mostly well written even though some sentences were occasionally confusing, I would comment careful reading of the whole manuscript. Some literature references could also be improved and updated. Some important considerations:

Comment 1: The annotation pipeline by Pineda et al. 2018 should be mentioned.

Comment 2: The authors only site Trinity for assembly and RSEM for quantification (lines 640-641). Please expand to include other softwares.

Comment 3: Figure 1, the authors write "Pulls expression information for each from toxin-like transcript.". This statement is confusing

Experimental design

Extracting useful information from transcriptomes is an important research question. The authors seemed to have fairly analysed their software but some additional analyses are required to fully understand if this pipeline could be useful at identifying toxins.

Comment 1: The authors should include a discussion on the inclusion of proteomics data on the identification of toxins. Transcriptomics data are stained with a range of sequencing errors and combining a transcriptome analysis with mass-spectrometry outputs has become a main stream method for identifying toxins in venoms. The authors do not seem to have considered this point in their pipeline and some comments on that point or how to integrate mass-spectrometry data with existing tools (protein pilot?) would be useful.

Comment 2: The authors should mention signalP as necessary programs in Figure 1.

Comment 3: It should also be mentioned how are partial sequences treated? Are they flagged or not as partial? Do they receive partial treatments?

Comment 4: The authors compared the number of toxins identified using their pipeline with the number of toxins identified in the literature for the analysis of the same transcriptomes. Could the author also analyse what is the overlap between their identified toxins and the one reported in the literature? ie are the toxins the same or novel?

Comment 5: What about when the reads do not need to be assembled? Cone snail toxin transcripts are small enough to be included in the raw reads of a 454 sequences, potentially removing problems with spurious assemblies created by Trinity and similar softwares. This should be mentioned also by the authors

Validity of the findings

In this research article the authors describe a new pipeline to analyse the toxin contents of a venom by analysing transcriptomics next generation sequencing data. Their pipeline combined the outputs of several programs that quantify expression level, identify open reading frames, signal sequences, closest sequences from a dataset (ToxProt) and create phylogenetic trees. They used their pipeline to analyse the transcriptomes of four next-generation sequencing datasets extracted from public repositories. The authors made a number of claims, which unfortunately do not seem to be fully supported by the data presented in the manuscript.

Claim 1: The authors claim that their method identified more candidate toxins than reported in the original analyses. The data presented by the authors do not support that clain. Indeed, Table 3 highlights that the pipeline developped by the authors has missed a large number of gene superfamilies in the cone snail transcriptome. Have the authors tried to identify what part of the pipeline was causing problems? For example, the con-ikot-ikot proteins are quite large and should be easy to identify. Why did the pipeline only returned 3 toxins from the T-superfamily whereas Phuong et al. 2016 discovered 56 toxins?

Claim 2: The authors claim that "Venomix is the first pipeline to provide all these outputs in an easy to use search strategy" but I cannot see any evidence of this claim, what is the format of the output that makes it so practical? I cannot see any summary tables being produced by the pipeline neither easy to understand.

Claim 3: The authors state that they provided a pipeline that would help researcher with little bioinformatics skills to analyse transcriptome data but they require the users to provide: assembled reads into transcriptome, gene expression information in a tab delimited output and a tab delimited blast output against the ToxProt database. This cannot be considered as user friendly unless the authors provide a protocle with their package to help non-bioinformatic researchers produce these files.

Reviewer 2 ·

Basic reporting

Macrander and colleagues propose a very interesting tool designed for researchers who are not specifcally into bioinformatic and want to isolate toxin candidates in high-troughput sequencing data. Such tool is of great interest for the scientific community and the way the pipeline has been designed perfectly fills the gap between a very basic similarity search protocole and a refined machine-learning (or other cutting-edge) based method.
While the protocole is well designed, the way it is presented through this article does not allow to understand the advantages of this tool. The text itself is ok, sometimes going into too much details though, but the major issues are figures and tables. The principale figure presenting the pipeline itself should be simplified to allow authors to catch in a glance what are the keys steps of the analysis. Tables which summarize the comparative analysis should be represented as graphs and all other figures should go into the supplementary material. The discussion is ok but the material and methods as well as the results parts have to be shorten.
I thus suggest authors to rewrite the manuscript focusing their attention first on how the protocole and results are illustrated and second on giving a more synthetic view of the results.

Experimental design

The pipeline is well design. These are juste suggestions to improve it :

1) If Venomix is designed to help non bioinformaticians in their search for new toxin candidates, I would like to see how it performs on a standard laptop, especially how long it takes for the similarity search. If the computing btime is reasonnable, it could be a big plus to mention it in the revised manuscript. Some people don’t have access to large cluster of calculation and it could encourage them to use Venomix
2) For the similarity search process you could probably improve the sensity using an hmm approach . Hmms profiles are easy to build.
3) You mentioned the influence of assembly steps. Something you could think about to adress this question is simpy mapping back the reads on toxin candidates (DNA sequences obviously) or even predicting ORF on short reads ( what people use in metagenomic studies).

Validity of the findings

Venomix seems to allow the identification of more toxin candidates which makes it a valuable tool for the scientific community. I would like to see some basic statisctics as T-tests to see more clearly if the pipeleine allows to retrieve more toxin candidates compared to what have been found in the original publications. I

Additional comments

Specific points to adressto be :

1) Key word : I think that transcriptome, toxins and bioinformatic are enough.
2) The term TMP has to be clearly define in the manuscript.
3) Figure 1 has to be modified a lot, mainly sinplified. Just draw some boxes with the major steps and remove all the remainig text. It was really hard forme to understand what the program was doing and how the decision to clasify a sequence as a toxin candidate was done. There are also some typos.
4) Table 1 has to go to supplementary and the commentaried about assembly parameters etc etc has to be shorten.
5) Line 760 : Do you mean toxins from distantly related taxonomic groups ?
6) Table 2 has to be drawn as a graphic. It would simply help the readers to understand how Venomix perform.
7) Line 765 : you just have to specify one time that the less stringent approach corresponds to the use of a higher e-value for the blast. A more general commentary about writing is that many concepts are named differently all along the manuscript. Sometines toxin candidates is quoted (line 791 and after), sometimes not. Sometimes the number of toxin is designed by N=x, sometimes the number is written before. This seems to be small details, but it would really help the readers to have allthis small little things cleaned.
8) Line 781 : too many o in Conodipin
9) Line 784 : something wrong with tha sentence ?
10) The description of each Venomix output is a little bit heavy, escpecially because they are not described systematicaly in the same order. You should make one big paragraph to summarize the results ans all the small details should go into the supplementary.
11) Line 805 ; ‘Have no known function’. Could you say based on what ?
12) Line 835 : Naive users have to know that rsem is a read mapping based approach as well as BWA. We should learn that at the begining of the manuscript, not at the end.
13) Table 3 and Table 4 should use the same exact same notation. Toxin family / super family ?
14) Line 846 and after : this is way too detailed.
15) Line 862 : what you named ‘no input’ are default parameters.

Reviewer 3 ·

Basic reporting

The manuscript appears to adhere to PeerJ policies and follows the standard PeerJ article template. It is written in English with clear unambiguous text.
In this manuscript, is presented Venomix as a new pipeline to characterize the toxin-like diversity from transcriptomes in phylogenetically distinct organisms. Venomix incoporates widely used programs into its pipeline and provides the user with several output files that can be used for different purposes. To evaluate the perfomance of Venomix, in this study were used four transcriptomes of different species. Venomix recovered numerically more candidate toxin transcripts for three of the four transcriptomes than the original analyses.

Undoubtedly this new tool is quite relevant in the area of "venomics" since can quickly identify toxin gene candidates from transcriptomes of a great variety of species. Although the authors note that Venomix is not designed to be used to as definitive validation pipeline fo toxin genes their outputs are very informative and can be used to further evaluate these candidates about several aspects.

Also, as the pipelines are versatile, mainly two parts of the workflow (sequence alignment and tree reconstruction) could easily be modified with the preferred programs which could give a comparative perspective within of this pipeline.

Experimental design

The methods are described in sufficient detail to be replicated. Research question is well defined and the knowledge gap is easily identifiable. Venomix succeeds to solve three problems found in similar approaches (Reproducibility, not be a toxic gene identifier and to accommodate a wide variety of taxa).
.

Validity of the findings

Overall, this study is clear from its original research until your conclusions being supported by your results. The authors meet the objetive to show that the Venomix package is a useful tool to identifiy and characterize the diversity of toxin-like transcripts.They also open a lot of possibilities in the development of pipelines centered around this goal, the identification of components of venoms in poorly studied lineages.

---

## Round 0.2 · accepted · Accept

Dear Dr. Macrander and colleagues:

Thanks for submitting your manuscript to PeerJ. I had hoped to have all of the original reviewers take a second look at your revision, but only one was available. Reviewer 3, as well as myself, believe that you have successfully entertained the concerns of all of the reviewers. Thus, I believe that your work is now acceptable for publication in PeerJ. Congratulations! I look forward to seeing this work in print, and I do expect Venomix to be a valuable tool for research on toxins. Thanks again for choosing PeerJ to publish such important work.

Best,

-joe

# Reviewer 3 ·

Basic reporting

The answers given to the other reviewers are adequate and the changes made to the manuscript have improved it. It is excellent to have this tool for research on venoms.

Experimental design

Experimental design is clear. The pipeline is well design. The modifications suggested in this part by the other reviewers were carried out appropriately.

Validity of the findings

The authors respond appropriately to the suggestions of the other reviewers. The authors make very clear the strengths and scope of the pipeline as well as its weaknesses and also where the pipeline could be optimized.

Additional comments

Congratulations for the work.